# Patterns of vascular access among chronic kidney disease patients on maintenance hemodialysis at Muhimbili National Hospital. A single centre cross-sectional study

**Daniel Msilanga**[1,2]\*, **Jacqueline Shoo**[1,2], **Jonathan Mngumi**[1,2]

**1** Renal Unit, Department of Internal Medicine, Muhimbili National Hospital, Dar es Salaam, Tanzania,
**2** School of Clinical Medicine, College of Medicine, Muhimbili University of Health and Allied Sciences, Dar es Salaam, Tanzania

\* pascodanny@gmail.com

## Abstract

Hemodialysis vascular access profoundly impacts the quality of care for chronic kidney disease (CKD) patients worldwide, with arteriovenous fistulas (AVFs) preferred for superior outcomes. Despite global guidelines, Sub-Saharan Africa, including Tanzania, faces challenges, by still relying on non-tunneled central venous catheters (CVCs) due to accessibility and financial constraints. We aimed to describe the pattern of vascular access use among CKD patients on maintenance hemodialysis at Muhimbili National Hospital. A cross-sectional study to describe the pattern of vascular access among patients with CKD on maintenance hemodialysis therapy. Descriptive statistics were used to summarize the baseline characteristics and patterns of vascular access. Our study received ethical clearance from the Muhimbili National Hospital Research Ethics Committee (Ref: MNH/IRB/VOL.1/2024/005). All consent forms were written and provided in English or Swahili. We analysed 200 study participants, with a mean age of 53.3 (14.5) years. Almost all participants initiated hemodialysis with nontunneled central venous catheters (95.5%). A substantial portion continued to use non-tunneled CVCs (25.5%) with mean duration of 7.1 (2.1) months, some transitioning to tunneled CVCs (39.5%) or AVFs (35%). Among patients with multiple nontunneled catheters, catheter dislodgement was the main indication for catheter replacement. Our study highlights the prevalent use of nontunneled central venous catheters (CVCs) as the primary vascular access method for CKD patients undergoing hemodialysis at Muhimbili National Hospital, Tanzania. These findings underscore the urgent need for analysis of the cost associated with non-tunneled catheter reliance and interventions to improve access to AVFs and enhance vascular access management, ultimately optimizing patient outcomes in resource-limited settings.

## Introduction

Chronic kidney disease (CKD) is a growing public health challenge in Tanzania, with an estimated 3% of the population affected by stage 5 CKD, requiring renal replacement therapy

**Data Availability Statement:** The dataset generated and/or analyzed during the current study is available as supplementary material.

**Funding:** We declare that this research was entirely self-funded. No external funding was used to support this work. The authors have no financial interests or relationships that could have influenced the study's design, data collection, analysis, interpretation, or manuscript preparation.

**Competing interests:** The authors have declared that no competing interests exist.

(RRT) [1]. As of 2023, access to hemodialysis was still limited, with fewer than 50 dialysis centers and only 649 hemodialysis machines nationwide, equating to 10 machines per million population [2]. Despite the rising CKD burden, Tanzania faces a shortage of trained personnel for dialysis vascular access creation, with fewer than 10 vascular surgeons to serve the entire population [2–4]. As a result, a significant number of CKD patients are forced to rely on un-recommended non-tunneled central venous catheters (CVCs) for hemodialysis, despite the well-documented risk of complications associated with long-term catheter use [3].

The Kidney Disease: Improving Global Outcomes (KDIGO) guidelines and the Fistula First Initiative advocate for arteriovenous fistulas (AVFs) as the preferred vascular access due to their superior long-term survival rates and lower risk of complications compared to catheters [5, 6]. However, in resource-limited settings like Tanzania, patients rely on central venous catheters due to late presentation with advanced CKD, requiring urgent dialysis and without prior access to predialysis care [7, 8]. Additionally, a lack of skilled personnel for advanced vascular access services further exacerbates reliance on non-tunneled CVCs [9–12]. This gap between global best practices and local conditions underscores the need for localized data on vascular access patterns, which remains limited [13].

The overarching goal of this study is to address this gap by describing the patterns of vascular access among CKD patients undergoing hemodialysis at the hospital. By examining current practices, our research aims to identify opportunities for improving vascular access management and enhancing the overall quality of care for CKD patients in Tanzania and similar settings.

## Methodology

### Ethics statement

Ethical clearance was obtained from Muhimbili National Hospital, Clinical Research, Training and Consultancy Unit with reference number MNH/IRB/VOL.1/2024/005. All participants provided written informed consent before any study procedures are conducted. The consent form documents were written and provided in English and Swahili language.

A cross-sectional study was conducted at the Muhimbili National Hospital (MNH) hemodialysis units from 10th April 2024 to 15th May 2024. The MNH is a largest tertiary-level public health facility with a 1500-bed capacity. The MNH has 50 hemodialysis machines and dialyzes between 100 and 130 CKD patients per day.

### Recruitment procedure

During each dialysis visit, patients were identified from the dialysis appointment registry and randomly selected using a rotary method. The study details, including the study's purpose, procedures, and potential risks and benefits, were explained to them, and they were invited to consent to participate in the study. Those who consented were included in the study. This process was repeated until the sample size was reached. Approximately half of the patients are covered by health insurance, whereas the remainder pay out-of-pocket for hemodialysis treatments. Due to financial constraints, almost all of those who are paying out-of-pocket are receiving dialysis therapy either twice or once weekly.

The data were collected through an interview-administered questionnaire. Selected clinical data, including comorbidities, frequency of hemodialysis treatments per week, and type of vascular access, were collected from the dialysis registry (a paper document). Vascular access type data included the type of initial access, anatomical location of vascular access, duration of access use, and current access, and to avoid the possibility of confusing dialysis access types, a physical examination of each patient was performed to ensure that proper vascular access was

recorded. The outcome was the pattern of vascular access use during the course of dialysis therapy.

## Data analysis

Descriptive statistics were employed to summarize the baseline characteristics of the study population and to outline the patterns of vascular access. Continuous variables were presented as means with standard deviations, while categorical variables were reported as frequencies and percentages.

## Authors access to study participant information and confidentiality

All information related to study participants will remain confidential and will be identifiable only by codes known to the researcher. To ensure participant privacy, all personal identifiers will be replaced with unique codes. Only the primary researcher will have access to the code key linking participants to their data. This process will safeguard participant confidentiality throughout the study.

## Results

Two hundred study participants were enrolled in this study. The mean age was 53.3 years (14.5), with almost half of the study participants being male (58.5%), unemployed (52.5%), under medical insurance (56%), and doing thrice weekly dialysis sessions (56.5%). Hypertension (46%) was the main etiology of CKD, followed by diabetes mellitus (30.5%) (Table 1).

**Table 1. Demographic data (n = 200).**

|  | N (%) |
|---|---|
| Mean age (years) | 53.3 (14.5) |
| Gender |  |
| Male | 117 (58.5) |
| Female | 83 (41.5) |
| Occupation |  |
| Unemployed | 105 (52.5) |
| Retired | 48 (24.0) |
| Employed | 47 (23.5) |
| Payment modality for HD |  |
| Insured | 112 (56.0) |
| Uninsured (out-of-pocket payment) | 88 (44.0) |
| Etiology of CKD |  |
| Hypertension | 92 (46.0) |
| DM | 61 (30.5) |
| HIV | 15 (7.5) |
| Glomerulonephritis | 14 (7.0) |
| Unknown cause | 13 (6.5) |
| Other causes | 5 (2.5) |
| Weekly frequency of hemodialysis sessions |  |
| Three times per week | 113 (56.5) |
| Two times per week | 80 (40) |
| Once per week | 7 (3.5) |

CKD: Chronic kidney disease, DM: Diabetes mellitus, HD: Hemodialysis

**Table 2. Type of hemodialysis access.**

| Type of Access | Initial Access (%) (N = 200) | | | Current access (%) (N = 200) | | |
|---|---|---|---|---|---|---|
| | Insured | Uninsured | Total | Insured | Uninsured | Total |
| Non-tunneled CVC | 104 (52) | 87 (43.5) | 191 (95.5) | 18 (9) | 33 (16.5) | 51 (25.5) |
| Tunneled CVC | 0 | 0 | 0 | 43 (21.5) | 36 (18) | 79 (39.5) |
| AVF | 8 (.04) | 1 (.005) | 9 (4.5) | 51 (25.5) | 19 (9.5) | 70 (35) |

CVC: Central venous catheter, AVF: arteriovenous fistula

A total of 95.5% of patients initiated hemodialysis using non-tunneled central venous catheters (CVCs) as their first access. This included 52% of insured patients and 43.5% of uninsured patients. Over time, there was a shift towards arteriovenous fistulas (AVFs) and tunneled CVCs. Among insured patients, 9% continued to use non-tunneled CVCs, 21% transitioned to tunneled CVCs, and 25.5% were using AVFs. In the uninsured group, 16.5% remained on non-tunneled CVCs, 18% switched to tunneled CVCs, and 9.5% had AVFs (Table 2).

Among the subset of prevalent ESRD patients who used only non-tunneled CVCs for dialysis (25.5%), the mean duration of non-tunneled CVC use was 7.1(2.1) months. A history of multiple non-tunneled CVC catheterizations was reported in 68.6% of patients, and dislodgement of the catheter was the most common reason for recatheterization Table 3.

## Discussion

Our study provides valuable insights into the status and vascular access practice patterns among patients with chronic kidney disease (CKD) undergoing maintenance hemodialysis therapy in Tanzania. Notably, nine out of ten study participants initiated hemodialysis therapy using non-tunneled catheters. Subsequently, three-quarters of these participants transitioned to either tunnelled catheters or arteriovenous fistulas (AVFs). Additionally, a quarter of the participants continued to use non-tunneled catheters as their permanent dialysis access, with an average duration of seven months.

Vascular access serves as the lifeline for patients undergoing dialysis therapy [14, 15]. We found that almost all of the participants were initiated on hemodialysis therapy using non-tunneled catheters. This trend mirrors findings reported in Kenya, Libya and Senegal, where more than 80% of incident CKD patients started hemodialysis therapy with a non-tunneled

**Table 3. Patients with nontunneled central venous catheters as the only access for hemodialysis (N = 51).**

| | N (%) |
|---|---|
| Mean duration of using non-tunneled CVC | 7.1 (2.1) months |
| Current site of nontunneled CVC | |
| Internal jugular vein | 38 (74.0) |
| Femoral vein | 13 (26.0) |
| History of using more than one nontunneled CVC since start of HD | |
| Yes | 35 (68.6) |
| No | 16 (31.4) |
| Mean non-tunneled CVC used per person | 2.8 |
| Causes of changing nontunneled CVC | |
| Catheter dislodgement | 17 |
| Purulent discharge from the catheter | 10 |
| Catheter clotted/poor blood flows | 8 |

catheter as the initial vascular access [16–18]. However, this is contrary to the practice in most higher income countries (HICs), where CKD patients start hemodialysis therapy either by using a tunnelled CVC and converting to an AVF or starting with an AVF [19, 20]. Despite the reported high rates of morbidity and mortality, non-tunnelled CVCs remain an important type of initial hemodialysis access for the majority CKD patients in LMICs [16, 21]. Likely contributors to this practice pattern include limited predialysis care, late presentation necessitating urgent hemodialysis and lack of expertise for tunneled CVCs or AVF [22, 23].

Nearly a quarter of patients continued to rely on non-tunneled catheters as permanent access, with the majority being those who paid out-of-pocket. These patients had a prolonged mean catheter use duration. Similar percentages were reported in Kenya and Senegal; however, the mean duration of catheter use in our setting exceeded theirs by three months [16, 17]. The chronic use of non-tunneled catheters may be driven by the high costs associated with tunneled catheters or arteriovenous fistula (AVF) creation, which can be more than five times the cost of non-tunneled catheters. While non-tunneled CVCs are intended as a temporary solution, their extended use raises significant concerns about increased risks of infection, thrombosis, and mechanical complications, leading to multiple hospital admissions and higher healthcare costs [14]. A number of those who chronically utilized non-tunneled CVCs had a history of multiple insertions. This highlights the challenges in maintaining vascular access with non-tunneled catheters, leading to higher overall healthcare resource utilization and costs, as well as an elevated cumulative risk of complications for patients.

Our study revealed the extensive reliance on non-tunneled central venous catheters (CVCs) for hemodialysis among chronic kidney disease (CKD) patients at Muhimbili National Hospital (MNH). As the largest tertiary referral center in Tanzania, MNH offers valuable insights into pre-dialysis and dialysis care in the country. However, the results are based on data from a single public facility, which presents limitations in terms of generalizing the findings to the broader population nationwide. Most CKD patients at MNH present with advanced disease requiring emergency hemodialysis. The findings highlight the challenges patients face in accessing recommended vascular access options, particularly arteriovenous fistulas (AVFs), in resource-limited settings.

Addressing vascular access challenges for CKD patients in Tanzania and similar settings requires both short- and long-term strategies. In the short term, focusing on training more general surgeons and urologists in arteriovenous fistula (AVF) creation is essential, along with expanding interventional radiology services to manage access complications. For long-term success, strengthening pre-dialysis care including permanent vascular access creation and establishing a sustainable healthcare financing model to cover high procedure costs are crucial. Collaboration with the government and NGOs to subsidize these efforts and policy changes to prioritize vascular access will help align Tanzania with global hemodialysis standards, ultimately improving patient outcomes.

## Limitation

Our study provides an overview of vascular access patterns among CKD patients on maintenance hemodialysis at Muhimbili National Hospital (MNH), acknowledging several limitations. Being conducted at a single public center, the findings may not be generalizable to private dialysis facilities or the wider population in Tanzania. Additionally, the study does not examine factors influencing the choice of vascular access types or their outcomes, such as infection rates or hospitalizations, hindering comprehensive conclusions about their impact on patient outcome. Despite these limitations, the findings reveal a significant reliance on non-tunneled central venous catheters (CVCs) for hemodialysis in low- and middle-income

countries (LMICs) like Tanzania, highlighting the need for further research into the costs and complications of prolonged CVC use to improve vascular access practices.

## Supporting information

**S1 Checklist. A completed STROBE checklist.**
(DOC)

**S1 Data. Dataset of hemodialysis patients.**
(XLSX)

## Acknowledgments

We thank the Muhimbili National Hospital for approving the research and extend our deep gratitude to the patients who participated in the study.

## Author Contributions

**Conceptualization:** Daniel Msilanga.

**Data curation:** Daniel Msilanga.

**Formal analysis:** Daniel Msilanga.

**Methodology:** Daniel Msilanga, Jacqueline Shoo, Jonathan Mngumi.

**Project administration:** Daniel Msilanga, Jonathan Mngumi.

**Supervision:** Jacqueline Shoo, Jonathan Mngumi.

**Writing – original draft:** Daniel Msilanga.

**Writing – review & editing:** Jacqueline Shoo, Jonathan Mngumi.

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
