## [Decision Letter · Decision Letter 0]

23 Sep 2024

PGPH-D-24-01604

Patterns of vascular access among chronic kidney disease patients on maintenance hemodialysis at Muhimbili National Hospital. A single centre cross-sectional study

Dear Dr. Msilanga,

Thank you for submitting your manuscript to PLOS Global Public Health. After careful consideration, we feel that it has merit but does not fully meet PLOS Global Public Health’s publication criteria as it currently stands. Therefore, we invite you to submit a revised version of the manuscript that addresses the points raised during the review process.

We look forward to receiving your revised manuscript.

Kind regards,

Valerie Ann Luyckx

Academic Editor

Additional Editor Comments (if provided):

The manuscript has been reviewed by 3 reviewers with varying perspectives. Overall the manuscript is of interest, but more detail is required on the context and on how this data can be used to move forwards and lead to positive change.

Reviewers' comments:

Reviewer's Responses to Questions

**Comments to the Author**

1. Does this manuscript meet PLOS Global Public Health’s publication criteria? Is the manuscript technically sound, and do the data support the conclusions? The manuscript must describe methodologically and ethically rigorous research with conclusions that are appropriately drawn based on the data presented.

Reviewer #1: Yes

Reviewer #2: Yes

Reviewer #3: Yes

2. Has the statistical analysis been performed appropriately and rigorously?

Reviewer #1: Yes

Reviewer #2: Yes

Reviewer #3: Yes

3. Have the authors made all data underlying the findings in their manuscript fully available (please refer to the Data Availability Statement at the start of the manuscript PDF file)?

Reviewer #1: Yes

Reviewer #2: Yes

Reviewer #3: Yes

4. Is the manuscript presented in an intelligible fashion and written in standard English?

Reviewer #1: Yes

Reviewer #2: No

Reviewer #3: Yes

5. Review Comments to the Author

Reviewer #1: The “Patterns of vascular access among chronic kidney disease patients on maintenance hemodialysis at Muhimbili National Hospital. A single centre cross-sectional study” is a well-written and original manuscript addressing an important and underexplored topic. Despite the well-known guidelines on how to better transition patients from predialysis care to hemodialysis, elective referral of patients with a functioning AVF is still a challenge in several locations. To study this theme, the authors designed a cross-sectional study involving 200 participants living in Tanzania. The descriptive results highlighted a high rate of nontunneled central venous catheters (CVCs) as the primary vascular access method for CKD patients undergoing hemodialysis at Muhimbili National Hospital, Tanzania. This data suggests the need to implement interventions to improve access to AVFs and enhance predialysis care.

- Are there public renal registries in Tanzania? Is it possible to estimate the number of patients in hemodialysis per million inhabitants and/or the number of patients living with CKD in the country?

- Can the authors provide further information concerning the number of dialysis centers, number of patients on hemodialysis, number of nephrologists and of vascular surgeons currently available in Tanzania?

- Can the authors elaborate on how the data collected at Muhimbili National Hospital could be further extrapolated to provide a broader understanding of the Tanzanian picture at predialysis/dialysis care?

- Can the survival rate of patients in Muhimbili National Hospital be compared to other centers where best practices in the transition to hemodialysis initiation and vascular access maintenance are in place?

- Are there any differences between the public and private sectors concerning predialysis care and vascular access management in Tanzania? Would it be possible to compare results from these sectors (e.g. mortality, prevalence of AVF, etc)?

- Considering the data presented in the manuscript to be a picture of the current practices in Tanzania, can the authors discuss in more detail what immediate measures could be put in place to change the ongoing practices in the short, middle and long terms? (e.g broad education programs, improved infrastructure, patient advocacy, universal public care, budget relocation, etc).

- Please check the spelling at line 92: (Data analysis: “escriptive” statistics were used).

Reviewer #2: The investigators discussed the importance of vascular access problems in a developing country. Although many guidelines strongly recommends fistular is the first (except elder patients) and recommends to do before starting dialysis, in real life it is not possible especially in developing countries. This study enlightened the importance of this again and showed where we are. I have few comments

1. A native speaker should check the manuscript and if possible an expert may help to revise their manuscript.

2. Although they only give descriptive information, Might be better to mention in more detail about analysis method.

3. The sample size is small but the findings are very interesting and very important, (%95!!!) almost all patients started dialysis with non tunnelled catheter (26% had femoral catheter). uninsured rate is very high!!! these patients had only 1 session/week.-Unfortunately, the sample size is very small and may not reflect the whole country.

4. I recommend the to add more centers to enlarge sample size, this will make your findings more meaningful.

5. the presentation of data is poor

6. it might be interesting to compare the outcome (infection, replacement, hospitalisation, mortality etc.) of patients with different vascular access types. This will help to make your findings more interesting.

Reviewer #3: good cross sectional study to identify challenges of vascular access in a LMIC public dialysis Centre . Maybe the authors could give recommendations on how to improve the Fistula first initiative , patients to start dialysis with AV fistula instead of a temporary line.

6. PLOS authors have the option to publish the peer review history of their article (what does this mean?). If published, this will include your full peer review and any attached files.

**Do you want your identity to be public for this peer review?** For information about this choice, including consent withdrawal, please see our Privacy Policy.

Reviewer #1: **Yes: **Augusto Cesar Soares dos Santos Junior

Reviewer #2: No

Reviewer #3: **Yes: **Prof Ahmed Sokwala

---

## [Decision Letter · Decision Letter 1]

31 Oct 2024

Patterns of vascular access among chronic kidney disease patients on maintenance hemodialysis at Muhimbili National Hospital. A single centre cross-sectional study

PGPH-D-24-01604R1

Dear Dr Msilanga,

We are pleased to inform you that your manuscript 'Patterns of vascular access among chronic kidney disease patients on maintenance hemodialysis at Muhimbili National Hospital. A single centre cross-sectional study' has been provisionally accepted for publication in PLOS Global Public Health.

Best regards,

Valerie Ann Luyckx

Academic Editor

Reviewer Comments (if any, and for reference):

Reviewer's Responses to Questions

**Comments to the Author**

1. If the authors have adequately addressed your comments raised in a previous round of review and you feel that this manuscript is now acceptable for publication, you may indicate that here to bypass the “Comments to the Author” section, enter your conflict of interest statement in the “Confidential to Editor” section, and submit your "Accept" recommendation.

Reviewer #1: All comments have been addressed

Reviewer #2: All comments have been addressed

2. Does this manuscript meet PLOS Global Public Health’s publication criteria? Is the manuscript technically sound, and do the data support the conclusions? The manuscript must describe methodologically and ethically rigorous research with conclusions that are appropriately drawn based on the data presented.

Reviewer #1: Yes

Reviewer #2: Partly

3. Has the statistical analysis been performed appropriately and rigorously?

Reviewer #1: Yes

Reviewer #2: Yes

4. Have the authors made all data underlying the findings in their manuscript fully available (please refer to the Data Availability Statement at the start of the manuscript PDF file)?

Reviewer #1: Yes

Reviewer #2: Yes

5. Is the manuscript presented in an intelligible fashion and written in standard English?

Reviewer #1: Yes

Reviewer #2: Yes

6. Review Comments to the Author

Reviewer #1: Thanks to the authors. All comments have been addressed.

Reviewer #2: The authors answered all my comments

7. PLOS authors have the option to publish the peer review history of their article (what does this mean?). If published, this will include your full peer review and any attached files.

**Do you want your identity to be public for this peer review?** For information about this choice, including consent withdrawal, please see our Privacy Policy.

Reviewer #1: **Yes: **Augusto Cesar Soares dos Santos Junior

Reviewer #2: **Yes: **Mehmet Kanbay
